# Therapeutic Impact of Gardasil^®^ in Recurrent Respiratory Papillomatosis: A Retrospective Study on RRP Patients

**DOI:** 10.3390/v17030321

**Published:** 2025-02-26

**Authors:** Jennifer Sieg, Asita Fazel, Elgar Susanne Quabius, Astrid Dempfle, Susanne Wiegand, Markus Hoffmann

**Affiliations:** 1Department of Otorhinolaryngology, Head and Neck Surgery, Christian-Albrechts University Kiel, 24105 Kiel, Germany; stu216439@mail.uni-kiel.de (J.S.); asita.fazel@uksh.de (A.F.); elgarsusanne.quabius@uksh.de (E.S.Q.); susanne.wiegand@uksh.de (S.W.); 2Institute of Medical Informatics and Statistics, Christian-Albrechts University Kiel, 24105 Kiel, Germany; dempfle@medinfo.uni-kiel.de

**Keywords:** HPV, RRP, vaccination, Gardasil

## Abstract

**Background:** Recurrent respiratory papillomatosis (RRP) is a rare, non-malignant disease caused by human papillomavirus (HPV) types 6 and 11. The condition primarily affects the larynx, potentially leading to life-threatening airway obstruction. It is more aggressive in younger patients, necessitating frequent surgical interventions. This study investigates the therapeutic potential of the prophylactic HPV vaccine Gardasil^®^ in RRP patients, focusing on its impact on lesion size and the frequency of surgical interventions. Furthermore, a literature review was conducted to analyze the factors influencing the decision to vaccinate. **Methods:** A retrospective analysis was conducted on 63 RRP patients treated from 2008 to 2021. Disease burden was assessed using the Derkay score and the annual frequency of laser-surgical ablations. Comparisons were made between pre- and post-vaccination periods in vaccinated patients (n = 18), and between first and second halves of the disease’s course in unvaccinated patients (n = 14). **Results:** A reduction in the frequency of surgical interventions post-vaccination (*p* < 0.05) could be seen. The cumulated Derkay score per year decreased after second and third vaccination (*p* < 0.05). The decision to be vaccinated is influenced by multiple factors (e.g., potential side-effects, sociocultural factors, impact of social media, pre-existing conditions and the wider context of the recent pandemic). **Conclusions:** Gardasil^®^ appears to reduce the frequency of surgery and lessen disease severity in RRP patients, supporting the potential role of HPV vaccination as a therapeutic option for RRP. Moreover, it is crucial to overcome skepticism towards vaccinations to prevent the development of HPV-associated diseases in the first place.

## 1. Introduction

Recurrent respiratory papillomatosis (RRP) is a rare non-malignant disease caused by infection with human papillomavirus (HPV), primarily low-risk HPV subtypes 6 and 11 [1]. Depending on the extent of the papillomas, patients can present with symptoms ranging from hoarseness to life-threatening shortness of breath. The disease is categorized as juvenile- (=early-onset; EO) or adult-onset (=late-onset; LO) based on whether RRP initially appears before or after the age of 12 [2]. There is a peak incidence in infancy and young to middle adulthood [3]. Clinical observations indicate a more aggressive course of disease in cases with an early age of onset [4], with a possible association with HPV11 subtype being discussed [1]. The male–female ratio in patients with EO-RRP is even, whereas LO-RRP is more prevalent in males [5]. It is difficult to ascertain the precise incidence of RRP. This discrepancy can be attributed to the varying methodologies employed in the collection and analysis of data (e.g., USA—national birth data vs. state-level data), resulting in varying results. Moreover, the RRP incidence has been reducing since the implementation of HPV vaccination programs for the main target group, particularly adolescent girls, from 2008 onwards, which consecutively has led to a significant reduction in transmission to children (e.g., EO-RRP incidence in Australia declined from 0.16 in 2012 to 0.022 per 100.000 in 2016) [5].

The therapeutic management of RRP remains challenging despite intensive scientific and clinical efforts to improve the patients’ course of disease. Patients often experience alternating time periods without the need for therapy, then intervals marked by frequent therapy-demanding recurrences in a short span of time. Due to the lack of successful alternative treatments such as radiotherapy, repeated surgical ablation is necessary to maintain voice quality and ensure secure airways. These procedures, typically performed under general anesthesia, use cold instruments or laser surgery. When surgical intervention alone is insufficient, adjuvant therapies are recommended, either intralesionally or systemically tailored to the individual course of the disease. Studies show that cidofovir and bevazizumab have become established off-label adjuvant treatments [6,7,8,9,10,11,12,13,14], but there are also reports on a positive effect of the administration of pembrolizumab [15,16,17]. In addition to the high treatment costs [18], the frequent recurrences and the associated repeated operations represent a challenge for the healthcare system, and justify efforts to control the disease by alternative means, such as the aforementioned therapy with biologics. Furthermore, RRP represents a considerable psychosocial burden for the patients [19]. A significant correlation between age, the frequency of surgery, and quality of life in patients with RRP could be demonstrated. Younger patients reported a higher prevalence of voice-related problems, while a higher frequency of operations was associated with upper stress levels [20]. RRP patients exhibit a markedly diminished voice-related quality of life and an increased prevalence of depressive symptoms [21].

Since the prophylactic HPV vaccination recommendation was introduced in Germany in 2007, clinical observations have given rise to the assumption that vaccinated patients experience a more favorable course of the disease compared to those who are not vaccinated within the vaccination recommendation. As a result, the off-label use of the vaccine has been recommended for RRP patients with severe disease. This study evaluates the course of RRP in patients to determine the effectiveness of prophylactic vaccination on reducing the frequency of recurrences and the spread of papillomas. The hypothesis is that patients with RRP demonstrate a favorable disease course following vaccination, characterized by a reduced incidence of surgical procedures and a lower Derkay score. Specifically, we conducted a retrospective analysis on the course of RRP in two study populations with an intraindividual comparison, analyzing the number of surgeries per year and the Derkay score in vaccinated patients before and after intervention, as well as between first and second halves of the disease’s course in unvaccinated patients.

## 2. Materials and Methods

### 2.1. Data Collection

To identify patients with RRP, we considered all cases diagnosed with benign neoplasm of the larynx (ICD-10 14.1) treated at the Dept. of Otorhinolaryngology, Head and Neck Surgery, University Hospital Schleswig-Holstein, Campus Kiel, Germany, between January 2008 and October 2021. The observation period started in 2008, following the HPV vaccine recommendation by the German Standing Committee on Vaccination in 2007. The study was approved by the local Ethics Committee (D 564/21).

Patient files documenting “papillomas” were analyzed using the hospital information system and archived patient files. A total of 63 patients were included in the study, while 7 cases were excluded due to incomplete documentation. The following parameters were collected: age, gender, pre-existing conditions, medication, previous operations, allergies, substance abuse, family history, social history, date of initial RRP diagnosis, adjuvant therapies for RRP, HPV vaccination status with date of vaccination, and complications during the disease course. Additionally, all outpatient visits were documented with corresponding symptoms (dysphonia, dyspnea, stridor, foreign body sensation, other) and diagnoses, as well as all surgical appointments, including histopathological findings such as HPV genotype.

### 2.2. Quantification of the Disease Burden

To objectively assess the disease burden, we considered two parameters—the average number of surgical interventions to remove papillomas per year and the severity of findings using the anatomical Derkay score [22]. Both procedures performed in our clinic and those performed in other clinics, as well as operations under general anesthesia and while awake, were documented and evaluated in the study. The frequency of surgical interventions was divided by each patient’s individual observation period in years, resulting in an average number of operations per year. To ensure accuracy in assessing disease burden, only patients observed for at least one year were included, avoiding distortion due to short observation time.

### 2.3. Assessment of Papilloma Expression by Derkay Score

The Derkay score is an evaluation system used for the objective assessment of the severity of the disease extent in RRP, which is based on a clinical and an anatomical component. In addition to the phonation status, the clinical component also includes the presence of stridor, the urgency of intervention and the extent of respiratory distress. In the study presented here, however, only the anatomical component was considered, as it is not possible to reliably evaluate the clinical component retrospectively. Derkay scores were determined on an anatomical basis for each operation based on the surgical reports by numerically categorizing the findings (0 = no lesion, 1 = flat lesion, 2 = raised lesion, 3 = complete involvement) for the following anatomical sites:(1)Laryngeal—lingual epiglottis, laryngeal epiglottis, anterior commissure, posterior commissure, aryepiglottic fold, false vocal fold, true vocal fold, arytenoid, subglottis;(2)Tracheal—upper third, middle third, lower third, main bronchus right, main bronchus left, tracheostoma;(3)Other localizations—nasal, palatal, pharyngeal, esophageal, pulmonary and others.

Paired structures were scored separately. Surgical controls without papillomas were not evaluated with the Derkay score to avoid a false lower score in the mean value for cases with more frequent microlaryngoscopies without ablation of a recurrence. If no recurrent papillomas were present in an observation period when considering the vaccination effect, the score was set to zero overall in order to be able to include the case in the evaluation, despite the lack of recurrence. An average value was then calculated from this data set for vaccinated patients in the period before and after each vaccination.

Two methods were used to generate the above-mentioned average value.

Derkay_Mean_: Primarily, the mean value of the Derkay scores of all ablations was calculated. This method has advantages in terms of simplicity of implementation and comprehensibility. In addition, there is a rather low risk of bias due to individual missing or undetectable scores. As the data are retrospective, information on the anatomical Derkay score may not be available in every report. In such cases, it is possible to use the available data and still obtain meaningful results without invalidating the entire analysis due to a few missing values or excluding individual patients. However, it is problematic that this method does not consider the number of surgical interventions performed. The average value of the score does not allow a distinction to be made between a patient who has only had one operation and a patient who has required several operations with the same Derkay score. For this reason, the mean value of the Derkay score can only be meaningfully used in combination with a statement about the number of interventions.

Derkay_Sum_: As an alternative average value, each score value was summed up and the result was divided by the observation period in years. In this way, an increased number of operations can be seen in a higher overall score, even if the papilloma extent remains identical. By referring to the observation period, it is possible to objectively compare periods of different durations. In addition to the complicated application, another disadvantage of this method is a higher bias if individual missing values are not replaced compared to the Derkay_Mean_. To remove this confounding factor, all patients with missing scores during the observation period were excluded from the Derkay_Sum_. As this method is associated with a very high workload, it was only analyzed for vaccinated patients.

### 2.4. Determination of the Observation Period and Inclusion Criteria for Assessing the Vaccination Effect

To ensure that the observation periods of the vaccinated patients before and after vaccination were the same, the longest possible follow-up before and after each vaccination was calculated individually for the vaccinated patients. The corresponding longest possible duration, which was available both before and after vaccination, was then defined as the observation period. This approach enables reliable comparisons and minimizes potential bias due to different observation periods before and after vaccination. The alternative method of comparing the period before the first vaccination with the period after the last vaccination for each patient was not possible due to the observation periods being too short or the number of patients included being too small. To assess the effect of the vaccination, the patients also had to have at least three ablations during the disease. The analysis was conducted for each vaccination (i.e., first, second, third) utilizing the longest possible observation period. Additionally, a distinct analysis was conducted where only one vaccination was considered for each patient, namely, the one with the longest observation period. Depending on the fulfilment of the inclusion criteria, patients could be included in different analyses, and thus considered more than once if necessary. For example, at the time of the third vaccination, if either the full one-year observation period had not yet been reached, or only the date of a single vaccination was known, these patients were only included in the groups of the first and second vaccinations.

For patients who had not received the vaccination, an observation period was defined for the first and second halves of the disease chronology. Subsequently, an analysis was conducted to ascertain whether there was a change in the number of operations per year and the Derkay score in the first half compared to the second half of the course of disease.

### 2.5. Statistical Analysis

The statistical analysis was performed using the software R (2022.07.2) and GraphPad Prism (version 9.5.1). The primary outcome measure was the number of ablations per year, the Derkay_Sum_ and Derkay_Mean_. The independent variable was the administration of the Gardasil vaccination. Missing values were not replaced and were not included in the analysis. Dichotomously distributed data were analyzed using Fisher’s exact test. Variables that were considered unpaired were analyzed by Welch test or Mann–Whitney U test, depending on the given condition. Paired analyses were performed using the T-test or Wilcoxon test, depending on the given condition. A *p*-value < 0.05 was considered statistically significant. Inclusion criteria differed depending on the aspects of the study being analyzed. Initially, all patients who had undergone papilloma removal were included for descriptive analysis of the cohort. Regarding the vaccination effect, only patients with a minimum observation period more than one year and at least three ablations were included. To eliminate potential confounding factors, patients with missing Derkay scores were excluded from analyses of the Derkay_Sum_.

## 3. Results

In total, 63 patients (21 female; 42 male) were included. The average age of onset was 41.1 ± 23.3 years (n = 61; Mdn = 44.3; Min = 0.8; Max = 82.8). Out of 63 patients, 13 got classified as juvenile-onset and 48/63 patients as adult-onset RRP. In 2 out of 63 patients, the date of initial diagnosis was not documented. Patients with a total observation period of at least one year were operated on 0.8 ± 0.8 (n = 43; Mdn = 0.5; Min = 0.04; Max = 3.8) times per year on average. The mean Derkay score was 3.5 ± 3.1 (Mdn = 2.3; Min = 1; Max = 15.3).

During the disease, 6/63 patients required a tracheotomy, with 3/6 patients in the context of a malignant transformation. Out of 63 patients, 3 received a systemic therapy with interferon. An intralesional injection of interferon was performed in 4/63 cases. In 12/63 patients, synechiae developed over the course of the disease. A topic application of mitomycin to reduce adhesions was enacted in 10/12 cases.

### 3.1. Description of the Subgroup of Vaccinated Patients

Out of 63 patients, 18 were vaccinated against the human papillomavirus (8 male, 10 female). In 5/18 vaccinated patients, no date of vaccination and no information on the vaccine used were available, such that these have been excluded from the consideration of the effect of the vaccination. The vaccine was administered by the following specialist departments: pediatrics (n = 1); ear, nose and throat medicine (n = 2); general medicine (n = 2); gynecology (n = 1). In 12/18 cases, it is unknown which department administered the vaccination. In one patient who was vaccinated at our center (n = 1), no side effects were documented, apart from slight redness and swelling at the injection site.

In 1 out of 18 cases, the age of RRP-onset was unknown. Of the remaining 17 vaccinated patients, 8 got classified as juvenile-onset and 9 as adult-onset RRP. The mean age of onset of all 17 vaccinated patients was 25.4 ± 22.2 years (Mdn = 18.9; Min = 0.8; Max = 66.2). The total mean duration of observation was 21.6 ± 20.2 years (n = 18; Mdn = 18.0; Min = 0.03; Max = 58.9). HPV 6 could be detected in 10 of 18 accordingly analyzed cases, and HPV11 in 4/18 cases. In 4/18 cases, the HPV subtype was not determined.

All three vaccination dates were available in 10 patients. Only two vaccination dates were available in 1/18 patients (case 2). In two other patients, only the dates of the first (case 13) or third vaccination (case 10) were available. At the time of first vaccination, the patients were 52.2 ± 20.4 years old on average (n = 12; Mdn = 58.96; Min = 15.6; Max = 77.6). The mean duration of disease at this time was 28.0 ± 22.9 years (n = 11; Mdn = 26.5; Min = 0.03; Max = 67.6).

### 3.2. Reduction of Surgical Interventions per Year After Each Vaccination

Regarding the operations per year, the median remains the same prior to post first vaccination, and decreases after the second and third vaccination (*p* < 0.05 in the one-tailed Wilcox test; Figure 1; Table 1). The decrease in the number of operations per year after vaccination is also shown as a trend in the mean value (Figure 1; Table 1).

### 3.3. Reduction in Derkay_Mean_ After Each Vaccination

There were no significant differences regarding the Derkay_Mean_ from prior to post vaccination (*p* > 0.05). However, a reduction in the extent of the papillomas after vaccination is also visible (Figure 2).

### 3.4. Reduction of Derkay_Sum_ After Each Vaccination

If all patients with missing single values of the score are excluded, there is a statistically significant reduction in the median in the one-tailed Wilcox test (*p* < 0.05) for Derkay_Sum_ regarding the second and third vaccination (Table 2; Figure 3).

### 3.5. Vaccination Effect in Each Vaccination with Longest Possible Observation Period per Case

A comparison of the data pertaining to the vaccination, with the longest possible observation period for each patient, reveals a decline in the number of ablations per year from a median of 1.7 (M = 1.8; SD = 1.0; Min = 0.5; Max = 4.2) to 0.3 (M = 0.7; SD = 0.8; Min = 0; Max = 2.1) (n = 11; *p* < 0.01 in the one-tailed Wilcox test). The Derkay_Mean_ also decreases from a median of 3 (M = 2.9; SD = 1.8; Min = 1; Max = 6.4) to 1 (M = 2.1; SD = 2.8; Min = 0; Max = 8). This difference is not statistically significant (*p* > 0.05). Regarding Derkay_Sum_, the median decreases from 5.2 (M = 6.1; SD = 4.7; Min = 1.3; Max = 12.7) to 3.0 (M = 2.6; SD = 3.1; Min = 0; Max = 9.5) (n = 9; *p* < 0.05 in the one-tailed Wilcox test) (Figure 4 and Figure 5).

### 3.6. Disease Burden of Unvaccinated Patients

A total of 14 unvaccinated patients of 63 patients overall was included in the study in case these fulfilled the inclusion criteria regarding the vaccination effect (observation period of at least one year and three ablations in the period mentioned). Of these 14 patients, 10 (71.4%) were male and 4 (28.6%) were female. In one patient (7.1%), the age at which RRP was first diagnosed was unclear. The average age of onset for the remaining 13 patients was 30.4 ± 18.1 years (Mdn = 34.1; Min = 6; Max = 55.1). Of these 13 patients, 3 developed the disease before the age of 12 (23.1%). The average total observation period was 22.4 ± 18.3 years (n = 14; Mdn = 20.1; Min = 1.5; Max = 62.7).

The frequency of ablations per year in the second half of the disease course (M = 0.5; SD = 0.7; Mdn = 0.3; Min = 0.1; Max = 2.7) remained consistent with the first half (M = 0.5; SD = 0.5; Mdn = 0.3; Min = 0.1; Max = 1.4) in the included cases (n = 14). The Derkay_Mean_ increased from 2.6 ± 1.6 (n = 9; Mdn = 2; Min = 1; Max = 5.7) to 4.8 ± 5.1 (Mdn = 1.5; Min = 1; Max = 14). A statistical analysis, employing the Wilcoxon test, revealed that there was no statistically significant difference in either parameter (*p* > 0.05; Figure 6).

## 4. Discussion

The pathophysiology of recurrent respiratory papillomatosis remains unclear. In addition to the possibility of the perinatal transmission of EO-RRP, a prenatal intrauterine infection cannot be discounted [23]. The detection of HPV DNA in the placenta [24] and amniotic fluid [25] supports the idea of a prenatal infection of the fetus. Assuming a primarily perinatal transmission of HPV infection in juvenile-onset RRP, it remains ambiguous as to why the incidence of RRP is rare compared to widespread vaginal colonization. The observation that immunodeficient patients are more prone to HPV-related illnesses than immunocompetent individuals indicates that patients with juvenile-onset RRP may exhibit an impaired immunological response [26,27]. For instance, a singular case report indicates the presence of an NLPR1 mutation, which is linked to heightened activity of the inflammasome [28]. Other studies indicate an increased expression of interleukins, impaired T-cell response and impaired vascular endothelial growth factor distribution [29,30,31]. It has been demonstrated that there is an absence of discernible antibodies to HPV types 6 and 11 in individuals who have been infected with HPV [32]. The hypothesis that the HPV vaccination stimulates antibody production to a greater extent than the infection itself is currently under investigation [33]. It can therefore be posited that the vaccination may prove an efficacious adjuvant therapy.

Due to the unclear pathophysiology and the limited therapeutic options, various adjuvant treatment trials have been carried out. Therapeutic attempts with interferon-α, photodynamic therapy and cidofovir were not finally successful in clinical use, and were controversial due to their various side effects [19]. On the other hand, the VEGF inhibitor bevacizumab is considered promising in studies, and is currently the preferred adjuvant therapy in clinical practice [6,7,8,9,10,11,12,13,14]. Furthermore, PD-L1 antibodies, such as Pembrolizumab, have been successfully used for HPV-associated diseases. Promising results have also been observed in the treatment of RRP [15,16,34,35,36]. The aforementioned adjuvant therapies, their clinical applications and associated adverse effects have been previously delineated in detail within the existing literature [19,37].

As regards prophylactic protection, three vaccines are currently available to prevent the human papillomavirus: Gardasil^®^, Gardasil^®^9 and Cervarix^®^. Gardasil^®^ was launched in October 2006 and targets the most common and dangerous strains of human papillomavirus (HPV) types 6, 11, 16 and 18. Gardasil^®^9 offers the most broad-spectrum coverage against nine different HPV strains. The efficacy and safety profiles of HPV vaccines are promising, especially if the vaccination is given before exposure to the virus [38,39]. The vaccines use virus-like particles (VLPs) of the main structure protein L1, and induce specific, neutralizing antibodies in high concentrations [40]. New vaccines are currently being developed (e.g., INO-3107 and PRGN 2012) that are explicitly intended for therapeutic use in RRP targeting the oncoproteins E6 and E7 to generate more robust T-cell responses [41,42]. In a clinical trial from 2024, PRGN-2012 showed a significant decrease in the median number of surgeries in one year from four prior to zero after treatment, with no serious side effects [42].

The results of the present study are supported by previous data. An analysis of 13 patients older than 45 years old showed a significant increase in the average intersurgical interval, from 126 ± 87 days pre-vaccination compared to 494 ± 588 days post-vaccination [43]. A previous meta-analysis by Goon et al. has also shown an overall reduction of 0.123 recurrences or surgeries per month from pre- to post-vaccination in 38 RRP-patients [44]. Another meta-analysis by Rosenberg et al. presented an extension of mean intersurgery intervals from 7.02 prior to 34.45 months post-vaccination in 63 RRP-patients [45]. Our present study additionally underlines the positive therapeutic effect of the prophylactic HPV vaccine Gardasil^®^ on the course of disease in patients with RRP, encompassing not only a reduction in the number of operations, but also a decrease in the recurrence burden in terms of tumor spread. The number of ablations per year decreased significantly post-vaccination, indicating a reduction in papilloma severity over time. Considering the ablations per year among vaccinated patients, the median decreased significantly after the second and third vaccinations (*p* < 0.05 in one-tailed Wilcox test; Table 1; Figure 1). This trend of declining ablations per year post-vaccination was also observed in the mean. However, there were no statistically significant differences in the Derkay_Mean_ (*p* > 0.05 in Wilcox test). Nevertheless, a reduction in papilloma spread post-vaccination was evident. In analyzing patients with complete scores, we see a statistically significant reduction in the Derkay_Sum_ over the observation period for the second and third vaccinations (*p* < 0.05 in one-tailed Wilcox test; Table 2; Figure 3). Similarly, considering each patient’s data from the vaccination with the longest observation period, there was a statistically significant decrease in ablations per year and Derkay_Sum_ (*p* < 0.01 in one-tailed Wilcox test; Figure 4 and Figure 5). This suggests a significant reduction in disease severity following vaccination. Furthermore, no evidence of a reduced disease burden in terms of fewer surgical interventions per year or a decreased Derkay score could be observed in patients who did not receive the HPV vaccination when comparing the first and the second halves of the disease’s course (Figure 6).

The limitations of the study include a lack of comprehensive documentation, a short observation period, and a possible bias in the determination of the Derkay score depending on the individually differing descriptive documentation of the various surgeons in the surgical report. In addition, the Derkay score was analyzed based on anatomical aspects without taking clinical aspects into account. Restricting the analysis of the Derkay_Sum_ to complete patient data only could also lead to distortions. Additionally, as already explained in the methods section, the comparison of the Derkay_Mean_ can only be interpreted in connection with the frequency of surgical interventions. The simultaneous administration of various adjuvant therapies makes it difficult to clearly attribute the positive effects to the vaccination. Furthermore, RRP is a rare disease associated with a small sample size leading to a low number of cases in this study. The disease is typified by marked variability in progression and the necessity for treatment, which has a considerable impact on the statistical analysis of the data. In particular, the considerable heterogeneity gives rise to a multitude of outliers, which exert a significant influence on the calculation of the variances and standard deviations. The heterogeneity of the data set is further compounded by the varying times of onset of the disease. Some patients present with symptoms as early as infancy, while others are only affected in old age, leading to different durations of observation, ranging from a few years to periods of 30 to 50 years. These significant deviations within the data set underscore the intrinsic variability of the disease and present a challenge for statistical standardization. Another important aspect to consider is the different composition of the vaccinated population compared to the total and unvaccinated populations, especially in terms of age and sex. In the vaccinated group, the proportion of female patients was 58.8% (10/17), compared with 34.4% (21/61) in the total population and 28.6% (4/14) in the unvaccinated group. In addition, 47.1% (8/17) of vaccinated patients had juvenile-onset disease, compared with 21.3% (13/61) in the total population and 21.4% (3/14) in the unvaccinated group. These differences may be due to the fact that juvenile patients tend to have a more aggressive course of the disease, and in clinical practice, an indication for vaccination is usually only given in cases with a high surgery rate and a large tumor mass. The high proportion of females raises the question of whether there is a gender-specific increase in disease activity in female patients. This should be investigated in larger cohorts. Due to the small number of cases, age and gender matching was not possible with our inclusion criteria. Future studies should aim to improve the comparability of the groups by using larger numbers of cases or prospective study designs to minimize such biases. In principle, the comparison between vaccinated and unvaccinated patients is methodologically challenging, as vaccinated patients often show a higher disease burden, making direct comparisons difficult. For unvaccinated patients, the observation period from the first presentation to the last follow-up examination was defined and divided into two phases in order to make the course of the disease comparable—the attempt to categorize the disease progression of unvaccinated patients into two distinct phases for enhanced visualization highlights the intrinsic limitations of this approach. To summarize, there is no practical methodological option for comparing vaccinated and unvaccinated patients due to differences in disease burden, temporal variability and the individual nature of disease progression. The number of operations per year was deliberately chosen as a key figure, as it is robust and comparable, and it minimizes distortions due to outliers, although it lacks insight into the temporal distribution of interventions. In order to simplify the analysis and minimize distortions caused by outliers, as the operation rate per year is a more robust and directly comparable key figure, this approach provides a more lucid and intelligible exposition for the reader. In comparison to the ISI, it fails to provide detailed insights into the temporal distribution of interventions, which can lead to the under-recognition of early changes, such as an extension of the surgical interval. Nevertheless, as the primary objective of this study did not encompass a detailed individual analysis, we concluded that this approach was the most appropriate, and it has previously been utilized in other studies.

The therapeutic effects of prophylactic HPV vaccination on RRP are rather unexpected in view of the vaccination approval in the early 2000s, as a benefit for existing disease was not foreseeable at the time. A hypothetical mechanistic explanation for this could be that it is not the remaining HPV-infected cells of the mucosa alone that proliferate and thus cause the recurrence, but that it may also be a case of new infections of previously uninfected neighboring infected cells. A potential therapeutic role of HPV vaccination has already been demonstrated in other HPV-associated lesions. A systematic review showed that both intralesional and systemic applications of HPV vaccines may offer advantages in the partial or complete regression of anogenital warts [46].

Furthermore, the positive effects of vaccination are consistent with the global decline in HPV-associated diseases with sufficient vaccination coverage. Nevertheless, actual vaccination rates remain surprisingly low due to factors such as socio-cultural variables, pre-existing conditions, government funding, age recommendations and vaccine availability, particularly in the context of the recently experienced pandemic. The significant positive effect of prophylactic HPV vaccination worldwide, combined with the negligible side effects of the vaccination, make the actual vaccination rates measured worldwide seem surprising. The high variability in vaccine uptake is due to a number of factors other than education and awareness. In the following, we present these in order to generate awareness of the fact that HPV-associated diseases will remain a common health issue despite available vaccination, and to encourage researchers to integrate these aspects into predictions of the vaccination-associated incidence of HPV-associated diseases.

### 4.1. Barriers to the Uptake of HPV Vaccination—Missing Education Regarding the Immunization

Parental opinion is the primary determinant of vaccine uptake, as children are usually vaccinated during their school years. Research identifies a lack of information, the young age of the child, and judgments about vaccine efficacy as common reasons for delayed vaccination, with the recommendation of a health care professional being the most important motivator for initiating the vaccination process [47].

Large regional differences in knowledge about HPV are described [5]. The main barriers are concerns about safety and side effects, while protection against cancer increases acceptance. A study conducted in France revealed that approximately one-third of the evaluated individuals had no prior awareness of HPV. In addition, 54% of respondents expressed concerns about possible side effects [48]. Similar findings were reported in surveys conducted in Poland, China, and Kenya, indicating that knowledge about HPV and willingness to be vaccinated are significantly influenced by the availability of information and trust in medical professionals [49,50,51]. In Serbia and Brazil, the primary motivations for vaccination were cited as the prevention of cancer and the protection against specific disease agents [52,53].

Worldwide, mothers tend to be more willing to vaccinate than fathers [5], which is also confirmed by a Japanese study [54]; the pivotal importance of recommendations is once again substantiated in this context. It is suggested that males tend to have less parental awareness [55]. Furthermore, the level of an individual’s educational achievement and their prior contact with the healthcare system are significant contributing factors. In Istanbul, gynecologists and pediatricians demonstrated a superior comprehension of the risk factors associated with cervical cancer, and female doctors exhibited a higher rate of vaccination of their children (29.4%) compared to other mothers (18.3%) [56]. A cross-sectional study conducted in the United States between 2017 and 2020 revealed that 40.4% of individuals without a high school diploma demonstrated awareness of HPV, compared to 78.2% of college graduates [57].

The literature reveals that targeted communication strategies, such as the utilization of inoculation theory (a socio-psychological model implicating that beliefs or attitudes can be reinforced by initially presenting individuals with weak counterarguments) and the enhancement of information dissemination, are fundamental in augmenting the willingness to be vaccinated [58]. The implementation of differentiated communication strategies is essential for effective engagement with diverse target groups [59]. The beneficial impact of implementing vaccination reminders [60], educational interventions [61] and web tools [62] can be seen in different studies. Various examinations show the importance of healthcare recommendation in vaccine-hesitant parents [63,64].

### 4.2. Barriers to the Uptake of HPV Vaccination—Vaccination Decision Influenced by Other Medical Conditions

Specific medical conditions have also been shown to influence willingness to perform the HPV vaccination [65]. An article from Sweden reported that people with mental health problems often have lower vaccination coverage due to impaired health perceptions and difficulties in accessing health services; participants diagnosed with a psychiatric disorder, autism or mental disability and prescribed antipsychotics had lower vaccination rates [66]. These findings are similar with those of an Australian study, which also exposed lower vaccination rates among girls with Down syndrome or autism [67]. Overall, the proposed explanations include distorted parental perceptions of risk and safety and social norms, which may influence vaccination status. Some parents refuse to vaccinate their children because of concerns such as the possible link between vaccines and autism, or fear of psychological harm to their children. Complacency about immunization and assumptions about the sexual behavior of children with disabilities were also identified as contributing factors. Additionally, access to vaccination may be hampered by the fact that children with illnesses often visit specialists who do not prioritize vaccination, or by absenteeism from school, which limits exposure to vaccination opportunities.

### 4.3. Barriers to the Uptake of HPV Vaccination—Influence by Adverse Childhood Experiences

Another study from the US examines the association between adverse childhood experiences (ACEs, e.g., emotional, physical or sexual abuse) and HPV vaccination among adults [68]. People with four or more ACEs were more likely to be vaccinated, according to data from 2019 to 2021. Some ACEs, including emotional abuse and a family history of mental illness, had particularly strong links to vaccine receipt. By 2022, the results were less significant, highlighting the need for long-term data collection.

### 4.4. Barriers to the Uptake of HPV Vaccination—Relevance of Sexual Orientation

It can be observed from several reports that there are variations in the vaccination rate between different groups depending on sexual orientation. Although overall vaccination rates remain relatively low, U.S. National Health Interview Survey data from 2013 to 2017 show that homosexual and bisexual individuals demonstrate a higher vaccination uptake than their heterosexual peers, irrespective of gender [69]. The same results were observed for homosexual and bisexual women during the period between 2009 and 2016 [70]. As with all other groups, there are barriers related to the high cost of treatment, fear of side effects and lack of information [71]. A Malaysian study conducted between 2019 and 2022 assessed the knowledge and attitudes of men who have sex with men about HPV vaccination [72]. A total of 60.3% expressed an intention to receive the vaccine. It was found that bisexual and homosexual participants were more willing to get vaccinated than heterosexual participants. The average knowledge level of the participants was 6.82 out of a total score of 13. High knowledge of HPV was significantly associated with willingness to be vaccinated. Despite recognizing that HPV poses serious health risks, participants reported low perceived HPV vaccination susceptibility. Recommendations from healthcare organizations and educational campaigns may also increase vaccine uptake among these groups. It is similarly evident that web-based programs are effective among young gay and bisexual men [73]. In addition, two-thirds of participants expected to be stigmatized in healthcare settings [72]. The findings emphasize the need to promote HPV awareness and risk perception among LGBTQ+, and to address issues of stigma and sensitization around HPV vaccination to increase vaccination coverage. It remains important to target these groups individually, as they are at higher risk of HPV-related cancers [74].

### 4.5. Barriers to the Uptake of HPV Vaccination—Influence of Religious Beliefs

The socio-cultural factors that may contribute to a decision against vaccination can also include the religious background of an individual.

A study of 367 female adolescents in England found that 89% were highly likely to accept the HPV vaccination, with Muslim women and members of Hindu/Sikh groups showing less willingness to be vaccinated [75]. Another review about Islamic countries showed that misperceptions, in particular about the link between the HPV vaccination and sexual activity, contribute to a reduced uptake [76]. A survey of 442 participants underlined that negative views were mainly reinforced by belief in sexual behavior norms; a slightly positive correlation between religiosity and willingness to vaccinate/knowledge thereof could be seen [77]. In fact, a survey among 307 female college students indicates that sexual activity is the primary factor influencing vaccination decisions [78].

A potential cause for this issue is the dearth of information and educational resources, possibly reinforced by limited accessibility within fixed religious structures. A study of 1403 Muslims in Oman found that two thirds were willing to be vaccinated against HPV, but only 25% were aware of HPV [79]. This discrepancy could underline the impact of healthcare recommendations even in the absence of personal expertise. A significant negative correlation between religiosity and various vaccination statistics could be seen in adolescent women. Participants who belonged to an organized religion demonstrated a lesser level of knowledge, and were additionally less likely to receive a vaccination recommendation from their physician [80]. An examination about the distinction between religiosity—defined as participation in religious social structures—and spirituality as a subjective commitment to spiritual or religious beliefs in relation to parents’ decisions regarding the HPV vaccination revealed a significant influence of religiosity on the discourse surrounding vaccination. Furthermore, the data indicate that spirituality is a significant factor influencing parents’ perceptions of their capacity to manage health concerns [81]. In Israel, a web-based program increased vaccination intention in different religious groups. However, this was not the case among the ultra-Orthodox community. Although there was an increase in knowledge among these participants, there was no change in vaccination decision, highlighting the challenge of engaging with individuals within the confines of fixed religious structures [82].

Relating to the Buddhist religion, a study carried out in Thailand has shown that parents who considered religion to be important had a higher vaccination acceptance rate [83]. Christian communities show a more contradictory and ambivalent picture. It could be demonstrated that only organizational religiosity (as opposed to intrinsic religiosity) was associated with the intention to seek information about HPV. The results pertaining to the uptake of vaccination in relation to religiosity exhibited a somewhat disparate pattern [84]. A survey of Catholic parents revealed a significantly higher likelihood of vaccination among Catholic parents compared to non-religious parents. Conversely, parents who frequently attended religious services demonstrated a greater propensity to refrain from vaccination compared to those who did not attend religious services [85].

In general, there is an association between religion and attitudes towards HPV vaccination that can be either positive or negative. The integration of religion into fixed religious frameworks appears to inhibit a more open-minded attitude and limit the dissemination of information, which in turn may result in lower vaccination rates. The assumption of a causal relationship between vaccination and the onset of sexual activity represents a significant barrier to vaccination. The authors of the cited literature highlight the importance of educational actions irrespective of religion. In this context, it is also evident that the involvement of religious leaders and groups in health education has the potential to improve acceptance due to their function as role models.

### 4.6. Barriers to the Uptake of HPV Vaccination—Influence of Social Media

The dissemination of information about the human papillomavirus vaccination is significantly influenced by social media. The prevalence of negative discourse and misinformation on these platforms is a cause for concern. An analysis of 1794 tweets in Scandinavia from 2019 revealed that a considerable number of tweets demonstrate a lack of confidence in vaccines, necessitating extensive information searches [86]. A further study of 2996 comments on HPV vaccination on social media identified common misinformation relating to adverse reactions, the necessity for vaccination, conspiracy theories and distrust of authority, particularly on Facebook. The data indicate that posts employing an educational or vaccine-critical approach were more likely to contain misinformation [87].

A survey of 452 parents revealed that confidence in vaccinations differs depending on media usage and political beliefs. A greater proportion of Democratic parents than Republican parents had their children immunized [88]. A survey of 1073 parents revealed that the presentation of positive, evidence-based messages was associated with an increased willingness to vaccinate their children. Conversely, the dissemination of negative or mixed information was linked to delays or refusals to vaccinate [89]. Furthermore, another examination hypothesized that an enhanced comprehension of HPV and a favorable attitude towards vaccines are linked with an increased willingness to be vaccinated, particularly among mothers. A significant positive correlation was observed between satisfaction with the information provided and the positive content of social media posts, leading to an increased willingness to be vaccinated [90]. Additionally, evidence-based messages that directly refute misinformation about the vaccine influenced parental attitudes and behavioral intentions positively. Information from trusted sources such as the Centers for Disease Control and Prevention was found to be particularly efficacious in increasing vaccination uptake [91].

### 4.7. Barriers to the Uptake of HPV Vaccination—Influence of COVID-19-Pandemic on the Use of Social Media

Overall, the pandemic has had a far-reaching impact regarding the increased usage of media, leading to the question of whether there is a causal relationship between increased social media use during the pandemic and a decline in vaccination rates.

A study on the utilization of health-related videos in the United States between 2017 and 2022 revealed an increase in the consumption of such content on the YouTube platform. Frequently searched topics were vaccines, HPV in men, and potential side effects [92]. A further analysis of a total of 653,000 tweets published after 2020 revealed that the most prevalent topics were side effects, personal anecdotes and vaccination obligations. In comparison to the pre-pandemic era, there has been a notable shift towards the presentation of more personal accounts of purported adverse effects associated with vaccination [93]. Furthermore, the discourse surrounding the HPV vaccination on Polish-language Facebook pages indicated that a small number of opinion leaders were responsible for most posts expressing negative sentiment towards the vaccine. Nevertheless, the overall level of interest in HPV vaccination remained relatively stable [94]. This could also be shown in an analysis of 596,987 tweets between January 2019 and May 2021 [95]. A vaccine-hesitant group predominantly articulated negative sentiments pertaining to vaccine safety, whereas a vaccine-ready group exhibited a predominantly neutral to positive disposition. With the advent of the COVID-19 pandemic, there was a notable decline in interest among the vaccine-ready cohort regarding HPV-related content on social media. However, despite this shift, the underlying messages remained largely unaltered. A review of the relationship between vaccine hesitancy for coronavirus vaccines and cancer vaccines, such as the human papillomavirus vaccine, remains inconclusive due to the limited availability of data and the paucity of studies explicitly investigating this topic [96].

A study of 138 HPV vaccination-related articles on Facebook between 2019 and 2021 even demonstrated a positive trend, with the proportion of non-critical articles increasing from 51% to 72%, and misinformation decreasing from 50% to 24% [97]. These findings suggest an improvement in the communication of positive vaccination messages, despite the persistence of misinformation in a quarter of the articles. Overall, parents were particularly fatigued by the abundance of controversial content, and sought definitive guidance from health organizations in the pandemic era [98].

## 5. Conclusions

In summary, a multitude of factors contribute to the decision to either accept or decline vaccination, including concerns about side effects, misinformation (amplified by social media), cultural and religious beliefs, and socio-economic barriers such as limited access. Addressing these concerns through targeted awareness campaigns and tailored strategies is crucial to improving vaccine uptake globally.

The use of Gardasil should be continued primarily in the preventive setting, as high vaccination rates of children before their first sexual contact led to a reduction in the number of HPV infection, as a result of which the RRP incidence is likely to decline. The present study has demonstrated a significant decrease in the number of excisions carried out annually following vaccination. Furthermore, there has been a notable reduction in papillomas in accordance with the Derkay score. The study did not demonstrate a decline in the disease burden among unvaccinated patients during the disease course. These findings highlight the value of vaccination in both preventive and therapeutic contexts. The continued use of Gardasil, alongside the development of new vaccines, offers promising prospects for managing HPV-related conditions, including recurrent respiratory papillomatosis. The use of Gardasil remains an option to be considered as a therapeutic intervention in RRP depending on the individual course of the disease and the patient’s wishes. All in all, RRP remains a challenging disease for both patients and clinicians.

## Figures and Tables

**Figure 1 viruses-17-00321-f001:**
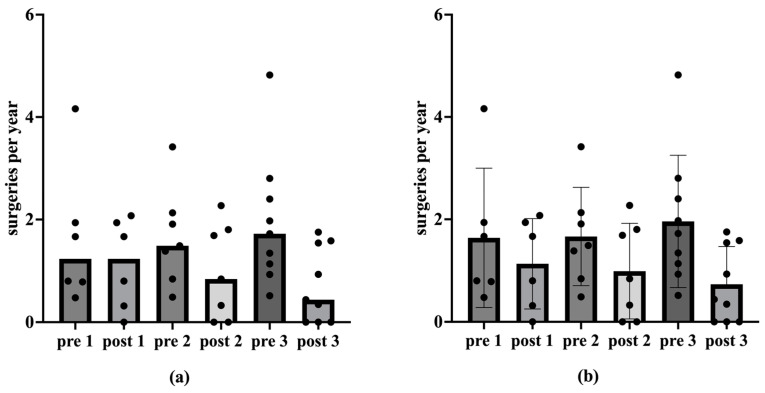
Medians, mean values and standard deviation of ablations per year before and after vaccination. The ablations per year are plotted on the ordinate. The abscissa shows the corresponding time before (pre) and after (post) vaccination. The numbers 1, 2 and 3 denote the vaccination under consideration. Figure (**a**) shows the median number of ablations per year before and after vaccination. The median remains statistically significant for the first vaccination (*p* = 0.047) at 1.24 ablations per year (n = 6) in the one-tailed Wilcox test. The one-tailed Wilcox test shows a statistically significant reduction in the median (*p* < 0.05) for the second vaccination from 1.49 to 0.84 (n = 7) and for the third vaccination from 1.72 to 0.44 (n = 9). Figure (**b**) shows the mean values and standard deviation of the ablations per year before and after vaccination.

**Figure 2 viruses-17-00321-f002:**
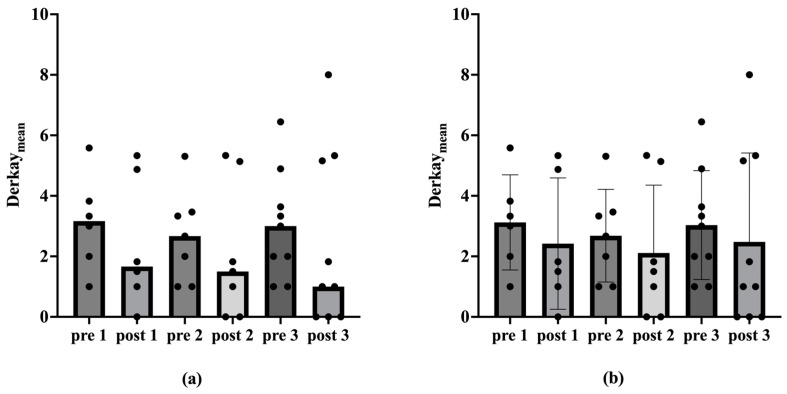
Medians, mean values and standard deviation of the Derkay_Mean_ before and after vaccination. The Derkay_Mean_ is plotted on the ordinate. The corresponding time point before (pre) and after (post) vaccination is shown on the abscissa. The numbers 1, 2 and 3 denote the vaccination under consideration. Figure (**a**) shows the medians of the Derkay_Mean_ before and after vaccination. Figure (**b**) shows the mean values and standard deviation of the Derkay_Mean_ before and after vaccination. The Wilcox test shows no statistically significant differences (*p* > 0.05).

**Figure 3 viruses-17-00321-f003:**
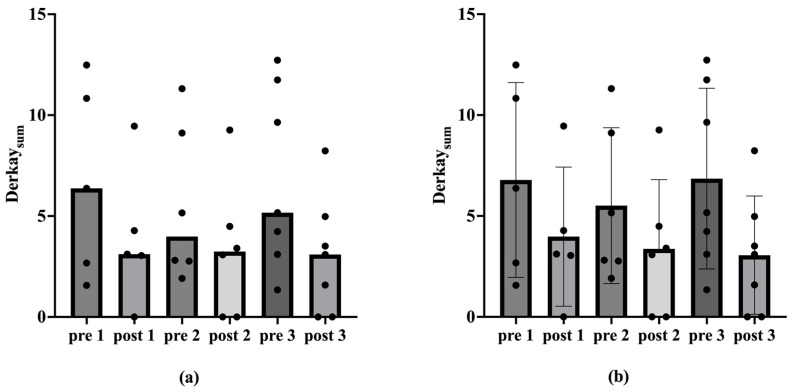
Medians, mean values and standard deviation of the Derkay_Sum_ before and after vaccination. The Derkay_Sum_ is plotted on the ordinate. The corresponding time point before (pre) and after (post) vaccination is shown on the abscissa. The numbers 1, 2 and 3 denote the vaccination under consideration. Figure (**a**) shows the medians of the Derkay_Sum_ before and after vaccination. Figure (**b**) shows the mean values and standard deviation of the Derkay_Sum_ before and after vaccination. The one-tailed Wilcox test shows a statistically significant reduction in the median (*p* < 0.05) for the second vaccination from 3.98 to 3.24 (n = 6) and for the third vaccination from 5.17 to 3.1 (n = 7).

**Figure 4 viruses-17-00321-f004:**
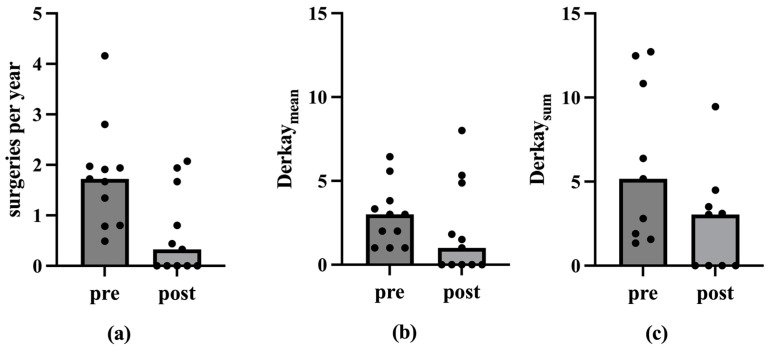
Medians of the ablations per year, the Derkay_Mean_ and the Derkay_Sum_ before and after vaccination with the longest possible observation period. The ordinate shows the ablations per year, the Derkay_Mean_ and the Derkay_Sum_. The abscissa shows the corresponding time before (pre) and after (post) vaccination (vac). Figure (**a**) shows the medians of the ablations per year before and after vaccination. The median decreases statistically significantly from 1.7 to 0.3 in the one-tailed Wilcox test (*p* < 0.01). Figure (**b**) shows the medians of the Derkay_Mean_ before and after vaccination. The median decreases from 3 to 1 (*p* > 0.05 in the Wilcox test). Figure (**c**) shows the medians of the Derkay_Sum_ before and after vaccination. The median decreases statistically significant from 5.2 to 3.0 in the one-tailed Wilcox test (*p* < 0.05).

**Figure 5 viruses-17-00321-f005:**
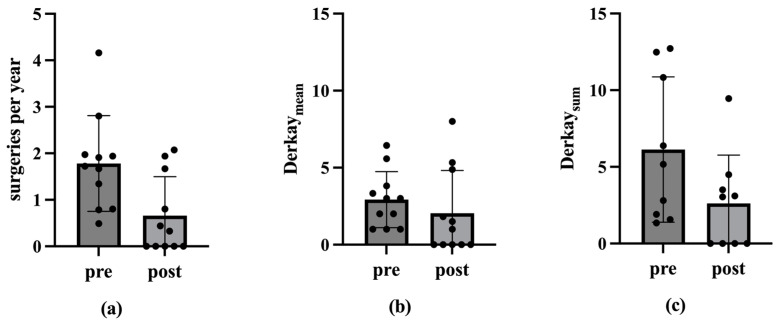
Mean values and standard deviation of the ablations per year, the Derkay_Mean_ and the Derkay_Sum_ before and after vaccination with the longest possible observation period. The ordinate shows the ablations per year, the Derkay_Mean_ and the Derkay_Sum_. The abscissa shows the corresponding time before (pre) and after (post) vaccination (vac). Figure (**a**) shows the mean values and standard deviation of the ablations per year before and after vaccination. Figure (**b**) shows the mean and standard deviation of the Derkay_Mean_ before and after vaccination. Figure (**c**) shows the mean values and standard deviation of the Derkay_Sum_ before and after vaccination.

**Figure 6 viruses-17-00321-f006:**
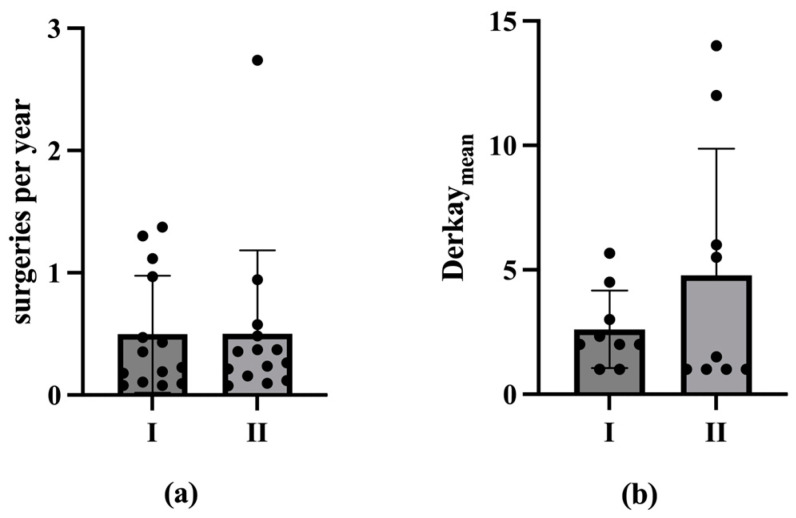
Burden of disease of unvaccinated patients compared between the first (I) and second (II) period of the disease. The mean and SD as well as the individual measured values are shown in both figures. (**a**) The frequency of ablations per year, plotted on the ordinate, of unvaccinated patients is compared between the first (I) and second (II) periods of the disease. The median changes from 0.31 in the first disease period to 0.29 in the second disease period without statistical significance (n = 14; *p* > 0.05 in the Wilcox test). In (**b**), the Derkay_Mean_, plotted on the ordinate, of unvaccinated patients is compared between the first (I) and second (II) periods of the disease. The median decreases from 2 in the first period of illness to 1.5 in the second period of illness without statistical significance (n = 9; *p* > 0.05 in the Wilcox test).

**Table 1 viruses-17-00321-t001:** Descriptive statistics of ablations per year before (pre) and after (post) first, second and third vaccination, as well as *p*-values of the corresponding one-tailed paired Wilcox test.

	Pre 1st	Post 1st	*p*-Value	Pre 2nd	Post 2nd	*p*-Value	Pre 3rd	Post 3rd	*p*-Value
number of patients	6	6	0.047	7	7	0.04	9	9	0.01
mean value	1.64	1.13		1.67	0.99		1.96	0.73	
standard deviation	1.36	0.88		0.96	0.93		1.29	0.73	
median value	1.24	1.24		1.49	0.84		1.72	0.44	
minimum	0.48	0		0.49	0		0.52	0	
maximum	4.16	2.08		3.42	2.27		4.82	1.75	

**Table 2 viruses-17-00321-t002:** Descriptive statistics of the summed Derkay score related to the observation period in years before (pre) and after (post) the first, second and third vaccination, as well as *p*-values of the respective one-tailed linked Wilcox test.

	Pre 1st	Post 1st	*p*-Value	Pre 2nd	Post 2nd	*p*-Value	Pre 3rd	Post 3rd	*p*-Value
number of patients	5	5	0.16	6	6	0.03	7	7	0.04
mean value	6.79	3.98		5.51	3.37		6.85	3.06	
standard deviation	4.83	3.45		3.86	3.43		4.48	2.93	
median value	6.38	3.11		3.98	3.24		5.17	3.1	
minimum	1.57	0		1.91	0		1.34	0	
maximum	12.49	9.46		11.31	9.2		12.72	8.32	

## Data Availability

The data supporting the findings of this study are available within the article. The anonymized data generated and analyzed during the current study are available upon reasonable request to qualified researchers.

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
