# Peer review of "Therapeutic Impact of Gardasil® in Recurrent Respiratory Papillomatosis: A Retrospective Study on RRP Patients"

_viruses, 2025, doi:10.3390/v17030321_

Round 1

Reviewer 1 Report

Comments and Suggestions for Authors

The article is interesting; however, I think that a comparison with other HPV-related diseases that may benefit from HPV vaccination should be at least cited in the discussion. Indeed, several recent studies described the vaccine's efficacy in treating ano-genital condylomas in both adults and children.
Consider for example:
Herzum A, Ciccarese G, Occella C, Gariazzo L, Pastorino C, Trave I, Viglizzo G. Treatment of Pediatric Anogenital Warts in the Era of HPV-Vaccine: A Literature Review. J Clin Med. 2023 Jun 23;12(13):4230. doi: 10.3390/jcm12134230. PMID: 37445264; PMCID: PMC10342328.
Villemure SE, Wilby KJ. A systematic review of the treatment of active anogenital warts with human papillomavirus vaccines. J Am Pharm Assoc (2003). 2024 Jan-Feb;64(1):179-185.e3. doi: 10.1016/j.japh.2023.10.028. Epub 2023 Oct 28. PMID: 38453661

Author Response

Response to Reviewer Comments

1. Summary

Dear Reviewer,

We are grateful for your consideration of this manuscript and for sharing your insights. The manuscript has been revised in line with your suggestions, with corresponding revisions highlighted in track changes in the re-submitted files. A detailed explanation can be found below.

Best regards,

Markus Hoffmann

2. Questions for General Evaluation

Reviewer’s Evaluation

Does the introduction provide sufficient background and include all relevant references?

Yes

Is the research design appropriate?

Yes

Are the methods adequately described?

Yes

Are the results clearly presented?

Yes

Are the conclusions supported by the results?

Can be improved

3. Point-by-point response to Comments and Suggestions for Authors

Comments 1: The article is interesting; however, I think that a comparison with other HPV-related diseases that may benefit from HPV vaccination should be at least cited in the discussion. Indeed, several recent studies described the vaccine's efficacy in treating anogenital condylomas in both adults and children.

Response 1: Thank you very much for your valuable comment! We have revised the discussion to include a comparison with other HPV-related diseases that may also benefit from vaccination (line 445 ff.). We have decided not to elaborate further on this topic as the discussion is already quite extensive. We fully understand the rationale and relevance of this topic, and there are indeed many sources on the subject. However, we feel it would be inappropriate to expand on this discussion in our manuscript as our discussion is already quite extensive. Nevertheless, we appreciate your thoughtful suggestion and will keep it in mind for future work.  

Reviewer 2 Report

Comments and Suggestions for Authors

This paper is two papers in one. The first half is a research paper and the second half a review of the literature on barriers to vaccination with HPV vaccines. I have no comments to make on the review half of the paper other than the “authors” on line 568 should have a reference. The first half of the paper investigates the therapeutic potential of the HPV vaccine Gardasil® in patients with recurrent respiratory papillomatosis (RRP), a disease caused by HPV types 6 and 11. ​ The research involved a retrospective analysis of 63 RRP patients treated between 2008 and 2021. ​ The study found that vaccination with Gardasil® significantly reduced the frequency of surgical interventions and the severity of papillomas, as measured by the Derkay score. ​ The results support the use of Gardasil® not only as a preventive measure but also as a therapeutic option for managing RRP. ​ The study highlights the importance of overcoming vaccination skepticism to prevent HPV-associated diseases.

One significant deficiency of this study is that the vaccinated population had a much higher proportion of women and juvenile onset patients compared to the overall population. Age at onset and sex are important factors for outcomes and it would have been a stronger study if they could have matched the vaccinated and unvaccinated patients on those characteristics. Since the study is retrospective it is impossible to correct this but it deserves discussion as to why this occurred and how this may have affected outcomes.

The most significant problem that I had with this paper was the number of patients in each group. In table 1/figure 1. There are 6 in P1, 7 in P2 and 9 in P3 however in the text above table 1 it referred to “all 17 patients”. Were the same patients in multiple groups? Needs clarity, as do the numbers in the other figures.

Minor comments:

Line 30 – The authors refer to RRP as a benign disease. Although RRP is not malignant the fact that so many surgeries are required suggests that it is hardly benign.

Line 40 – Rewrite sentence.

Line 73 – Sentence starting “The aim of this study…    In science we either test hypotheses, make observations or collect data on some disease or other phenomena. Saying that you want to demonstrate suggests that you are biased in your approach. Suggest leaving out.

Author Response

Response to Reviewer Comments

1. Summary

Dear Reviewer,

We are grateful for your consideration of this manuscript and for sharing your insights. The manuscript has been extensively revised in line with your suggestions, with corresponding revisions highlighted in track changes in the re-submitted files. A detailed explanation of the individual points can be found below.

Best regards,

Markus Hoffmann

2. Questions for General Evaluation

Reviewer’s Evaluation

Does the introduction provide sufficient background and include all relevant references?

Yes

Is the research design appropriate?

Yes

Are the methods adequately described?

Can be improved

Are the results clearly presented?

Can be improved

Are the conclusions supported by the results?

Yes

3. Point-by-point response to Comments and Suggestions for Authors

Comments 1:  This paper is two papers in one. The first half is a research paper and the second half a review of the literature on barriers to vaccination with HPV vaccines. I have no comments to make on the review half of the paper other than the “authors” on line 568 should have a reference.

Response 1: Thank you for your feedback. We were referring to all the authors of the previous literature in general, so there is no single reference. We have adjusted the section accordingly so that there are no more misunderstandings (line 591).

Comments 2:  The first half of the paper investigates the therapeutic potential of the HPV vaccine Gardasil® in patients with recurrent respiratory papillomatosis (RRP), a disease caused by HPV types 6 and 11. One significant deficiency of this study is that the vaccinated population had a much higher proportion of women and juvenile onset patients compared to the overall population. Age at onset and sex are important factors for outcomes and it would have been a stronger study if they could have matched the vaccinated and unvaccinated patients on those characteristics. Since the study is retrospective, it is impossible to correct this but it deserves discussion as to why this occurred and how this may have affected outcomes.

Response 2: We would like to sincerely thank you for this important and insightful aspect that we have added to the discussion (line 406ff.).

Comments 3:  The most significant problem that I had with this paper was the number of patients in each group. In table 1/figure 1. There are 6 in P1, 7 in P2 and 9 in P3 however in the text above table 1 it referred to “all 17 patients”. Were the same patients in multiple groups? Needs clarity, as do the numbers in the other figures.

Response 3:  Thank you very much for pointing this out. We clarified this issue in the methods (line 162ff.).

Comments 4:  Line 30 – The authors refer to RRP as a benign disease. Although RRP is not malignant the fact that so many surgeries are required suggests that it is hardly benign. Line 40 – Rewrite sentence. Line 73 – Sentence starting “The aim of this study…    In science we either test hypotheses, make observations or collect data on some disease or other phenomena. Saying that you want to demonstrate suggests that you are biased in your approach. Suggest leaving out.

Response 4: In this context, the term 'benign' is employed to denote the histopathological aspect of the dignity of the mass. To facilitate comprehension for the reader, the term 'non-malignant' has been utilized as a more accessible alternative. Furthermore, we have clarified the sentence in line 40 as requested and removed the sentence in line 73 as suggested.

Round 2

Reviewer 1 Report

Comments and Suggestions for Authors

Suitable for publication.